# Simultaneous Hip and Distal Radius Fractures—Does It Make a Difference with Respect to Rehabilitation?

**DOI:** 10.3390/geriatrics4040066

**Published:** 2019-11-28

**Authors:** Emilija Dubljanin-Raspopović, Marković-Denić Lj, Marko Kadija, Sanja Tomanović Vujadinović, Goran Tulić, Ivan Selaković, Milica Aleksić

**Affiliations:** 1Clinic for Physical Medicine and Rehabilitation, Clinical Center Serbia, 11000 Belgrade, Serbia; drsanjatv@gmail.com (S.T.V.); ivan.selakovic@gmail.com (I.S.); 2Faculty of Medicine, University of Belgrade, 11000 Belgrade, Serbia; kadija.marko@gmail.com (M.K.); ditulio@gmail.com (G.T.); aleksic.milica@yahoo.com (M.A.); 3Institute of epidemiology, Faculty of Medicine, University of Belgrade, 11000 Belgrade, Serbia; markovic.denic@gmail.com; 4Clinic for Orthopaedic Surgery and Traumatology, Clinical Center Serbia, 11000 Belgrade, Serbia

**Keywords:** hip fracture, wrist fracture, rehabilitation, functional outcome, mortality

## Abstract

Introduction: A minority of patients with hip fractures sustain concomitant wrist fractures. Little is known about the rehabilitation outcome in this group of patients. Aim of study: Prospective investigation of functional outcome and survival in patients with combined hip and wrist fractures compared with patients who sustain an isolated hip fracture. Methods: 341 patients who presented with an acute hip fracture during a 12 month period were included in the study. Outcome at discharge and 4 months follow-up was compared between patients with isolated hip fractures and those patients who sustained simultaneous distal wrist fractures. Results: The actual incidence of concurrent hip and wrist fractures in our cohort was 4.7%. Patients who sustained a concurrent hip and wrist fracture showed no differences regarding short- and long-term functional outcome and survival. Conclusion: Our results imply that patients with simultaneous hip and wrist fractures have no difference in rehabilitative outcome. Future studies should further investigate the distinctive characteristics of this patient subgroup.

## 1. Introduction

Hip fractures are a serious injury associated with significant morbidity and mortality. A minority of patients with hip fractures sustain concomitant wrist fractures. However, given the incidence of hip fractures, this number is not insignificant. This combination involves a greater force than in isolated hip fracture and is probably more traumatic to the patients. It is appealing to assume that the combination fracture is indicative of a frailer patient and poses a greater risk to life. Rehabilitation of patients with concurrent wrist fractures represents an additional challenge in this frail population. There is a clear gap regarding several aspects in the present literature on this topic. The limited amount of evidence revealing the rehabilitation potential of patients with concurrent hip and wrist fractures caused by a single fall is derived only from retrospective studies [1,2,3,4,5]. In addition, studies published on this topic focus primarily on survival, while functional outcome in this subgroup of patients is investigated to a lesser extent [1,3,5]. Also, the majority of studies analyze all upper extremity fractures together, while very few studies address specifically distal radius fracture in combination with hip fracture. The results of the existing studies are conflicting as well. While some studies report longer hospital stay [6], higher in-hospital mortality rates [2,6], and a less likelihood to be discharged home [6], others show favorable outcomes in this group of patients [1,2,4,5,7]. The aim of our study is to prospectively compare functional outcomes and survival between patients with combined hip and wrist fractures and those with isolated hip fractures. 

## 2. Methods

We examined 341 patients who presented to a university-associated orthopedic hospital in Serbia with acute hip fractures during a 12 month period. We included hip fracture patients with a concomitant wrist fracture obtained from the same injury, but excluded those with other simultaneous fractures or dislocations. We also excluded patients under the age of 65 years, patients with subtrochanteric and pathologic fractures, second ipsilateral hip fractures, major concomitant injuries, multiple trauma, malignant diseases, imminent death as a result of an end-stage disease, inability to walk before fracture, and nonoperative treatment resulting from high surgical risk. Three hundred and forty-one patients (258 females; 79.4%) met the inclusion criteria, and were enrolled in an open, prospective, observational cohort study. All patients with simultaneous wrist fractures were managed nonoperatively with cast immobilization after reduction of the fracture under anesthesia.

In order to compare the outcomes between patients with isolated hip fracture and those with a concomitant wrist fracture, the study population was divided in two groups. The first group (*n* = 325; 95.3%) included those patients who sustained an isolated hip fracture, while the second group (*n* = 16; 4.7%) included those patients who presented with a combined hip and wrist fracture.

We collected data concerning sociodemographic variables (age, sex, residential status), cognitive level, and prefracture functional status on admission using a standard patient or proxy interview. Home or residential status was defined as living in own home or in an institution. Waiting time for surgery, surgical risk, type of fracture, type of anesthesia, functional outcome at discharge, presence of postoperative complications, and length of stay (LOS) were recorded during the acute hospital stay. Patients who died during their acute hospital stay were excluded from the length of stay assessment. All assessments were performed by one tester who was not involved in the treatment of the patients, except for the American Society of Anesthesiologists (ASA) physical status classification of surgical risk and the type of fracture, which were classified by the attending anesthesiologists and surgeons, respectively. The Short Portable Mental Status Questionnaire (SPMSQ) was used to assess cognitive level at admission [8]. The 10-item questionnaire classifies the patient’s cognitive level as lucid (score 8–10), mild to moderate cognitive dysfunction (3–7), and severe cognitive dysfunction (0–2) depending on the number of correct answers. In patients with an SPMSQ score <3, all observed variables, except for the cognitive level, were collected from a proxy. The motor subscale of the Functional Independence Measure (FIM) that rates the patient independence from 1 (total assistance) to 7 (complete independence) was used to measure pre-fracture functional status and functional status at discharge and four months follow-up [9]. The motor FIM scale is comprised of 13 items and rates the patient independence in self-care (feeding, grooming, bathing, dressing upper and lower body, toileting), sphincter control (bladder management and bowel management), transfer (bed, chair, wheelchair transfer, toilet, and tub or shower transfer), and locomotion (walking, climbing stairs). Instrumental activities of daily living (IADL) prior to fracture and follow-up were assessed using the IADL scale [10]. This scale assesses eight skills, which are considered more complex than the basic activities of daily living. A summary score ranges from 0 (low function, dependent) to 8 (high function, independent). The ASA rating of operative risk was used to group patients’ physical level into one of five categories, ranging from 1 (healthy) to 5 (moribund) [11]. For the purpose of this study, ASA classes 1 and 2, and ASA classes 3 and 4 were combined. This approach was already used in previous studies [12,13]. 

No patient in our study was graded as moribund. All patients with intracapsular fractures (202 patients; 62.2%) underwent bipolar hemiarthroplasty, whereas open reduction and internal fixation with a dynamic hip screw was performed in all patients with extracapsular fractures (123; 37.8%). In all patients, depending on overall postoperative health status, early assisted ambulation was encouraged on the first postoperative day with weight bearing as tolerated, and all patients followed a standardized postoperative rehabilitation program. 

The two investigated groups were compared at baseline, discharge, and at 4 months follow-up. At discharge, they were compared regarding functional recovery, in-hospital mortality, presence of postoperative medical complications, and LOS. Observed postoperative medical complications were pneumonia, pulmonary embolism, delirium, deep venous thrombosis (DVT), urinary tract infection (UTI), deep wound infection, pressure sores, and prosthetic dislocation. Delirium was assessed using the Confusion Assessment Method on a daily basis [14]. After 4 months, the two groups were compared regarding functional independence (motor FIM and IADL scale) and mortality. Functional independence was expressed as the absolute motor FIM gain, which is the difference between 4 month follow-up motor FIM and discharge motor FIM. All variables were assessed by telephone interview. 

Continuous variables are presented in terms of mean values with SD and median with IQR. Categorical values are summarized as absolute frequencies and percentages. Continuous variables between groups were tested by the Mann–Whitney U test, while Chi-square test was used to test categorical variables that were expressed as numbers and percentages of patients. 

SPSS version 21.0 was used for statistical analyses. All analyses used two-tailed significance level of *p* < 0.05.

We obtained approval from the University’s institutional review board (tracking number 440/III-8).

On admission, all patients or their caregivers gave written consent to participate in the study.

## 3. Results

In our cohort, 16 patients (4.7%) sustained a simultaneous wrist and hip fracture. All wrist fractures were ipsilateral. There was no statistically significant difference regarding any baseline variable in our cohort (Table 1, Table 2 and Table 3). 

During the acute hospital stay, 23 (7.1%) had died. At discharge, patients with a combined hip and wrist fracture did not show statistically significant difference regarding LOS, complication rate, mortality, and functional recovery, measured with the motor FIM score. 

At 4 months follow-up, 47 (14.5%) had died. There was no statistically significant difference between the two groups regarding any observed outcome variable. 

## 4. Conclusions

This study investigated the effect of combined hip and wrist fractures on rehabilitation outcomes. Wrist and hip fractures are two of the most common fractures treated by the orthopedic surgeon. The actual incidence of concurrent hip and wrist fractures in our cohort was 4.7%, which is higher compared with Tow et al’s (2.6%) [1], Robinson et al al’s (1.8%) [4], Uzoigwe et al. (1.7%) [2], and Mullhal et al’s. (3.7%) [7] series. The higher percentage of concurrent wrist fractures in our cohort is a result of generally insufficient treatment of osteoporosis and consequently, ineffective prevention of fractures in the geriatric population in our clinical setting [15]. All fractures in our cohort were ipsilateral. The predominance of ipsilateral wrist fractures was confirmed in literature [2] and can be explained by the fact that in elderly patients, even a minimal transmission of force from the outstretched hand to the osteoporotic hip can cause a fracture [3].

We found no statistically significant difference between patients who sustained a concurrent hip and wrist fracture regarding any baseline variable. Moreover, our study revealed no difference regarding any outcome measure between the two observed groups at discharge and 4 months postoperatively. Previous studies reported a higher proportion of females in the multiply injured group [2,6]

Most authors investigating survivorship in patients with combined upper extremity and hip fractures showed no statistically significant increased short- and long-term mortality when compared with patients with isolated hip fracture [1,2,4,7]. Moreover, two groups of authors showed a trend towards lower 1 year mortality for patients with a simultaneous wrist fracture [2,4]. However, Thayer et al., who investigated concomitant upper extremity fractures in older people with hip fracture, found a higher risk of in-hospital mortality rates [6]. 

Patients with simultaneous wrist and hip fractures did not have longer lengths of hospital stay in our study. Monaco et al. also showed that LOS was not significantly influenced by the upper limb fracture [16]. However, other authors reported that the length of hospital stay was significantly longer for those with a concomitant upper limb fracture [1,2,4,6]. We argue that the length of hospital stay is not necessarily an objective outcome measure in our setting because most of our patients are directly discharged to rehabilitation hospitals, whereby the time of discharge is often dependent on the capacity of the facilities.

Patients with a hip fracture are postoperatively typically mobilized with a frame or crutches. However, patients with a simultaneous fracture who have a Coles cast can only be immobilized in a walking frame with elbow supports. Nevertheless, patients with a simultaneous wrist fracture did not show a lower level of functional independence at discharge compared with patients with an isolated hip fracture in our cohort. Our study also revealed that hip fracture patients with a concurrent wrist fracture did not have worse functional outcome at 4 months follow-up regarding ADL and IADL. Monaco et al. also revealed that concomitant fractures of the upper limb are not associated with the Barthel index at discharge from the acute setting and after rehabilitation [5]. Shabat et al. reported functional outcome after discharge, concluding that the majority of patients in both investigated groups returned to their previous ADL, but they failed to use a specific outcome measure to quantify the degree of functional recovery [3]. In contrast to these results, two groups of authors revealed impaired early ambulatory status, believing that the confluent upper limb injury explains this finding [1,17]. 

The successful prognosis in terms of survival and functional outcome at discharge and 4 months follow-up in patients with combined trauma revealed in our study implies that those patients do not have a lower rehabilitative potential despite the initially more severe trauma. Robinson et al. suppose that patients who sustain a combined wrist fracture have better preserved protective reflexes [4]. Similarly, Shabat et al. believe that these old people are more alert and have a better response to trauma then expected at their age, which can contribute to better outcomes [3]. Two other groups of authors argue that patients who suffer a simultaneous wrist fracture tended to be fitter prior to injury [1,4]. Cummings and Nevitte believe that patients who have a faster gait are more likely to fall forwards and sustain an upper limb fracture when a fall occurs [18]. Therefore, they consequently attempt to break their fall with the upper limb, thus sustaining simultaneous fractures [1,4]. Recent studies show that walking speed is a single good estimator of frailty, and is even considered to be a vital indicator [19]. In light of this, we argue that patients who sustain a simultaneous wrist and hip fracture have a lower level of frailty, and are thus less vulnerable to an initially greater amount of trauma. 

Our study has several positive points. First, all the previously published studies have a retrospective design. Moreover, there are just a few studies that specifically analyze distal radius fractures in combination with hip fracture, while most of them group all upper extremity fractures together. To the best of our knowledge, this is the first prospective patient series to analyze survival and functional recovery in patients with simultaneous wrist and hip fractures, thus minimizing the bias associated with previous retrospective studies. Furthermore, this is the first study to precisely evaluate functional outcome after discharge in this specific group of patients. Shabat et al. were the only ones to report functional outcome after discharge, but did not use specific outcome measures to quantify the degree of recovery [3]. The major limitation of the current study is the relatively small number of patients with concomitant fractures and the consequent failure to reach statistical significance for some aspects of comparison, which is an intrinsic problem of small cohort studies. This could be overcome with longer duration or multicenter design of further prospective studies investigating this topic. Furthermore, we did not study if surgical treatment of concurrent wrist fractures facilitates better early functional outcome. Recent studies suggest that internal fixation of distal radial fractures with volar locking plates eases early mobilization and as such, is being advocated even for elderly patients [20]. 

Despite the initially more traumatic event, patients who sustain simultaneous hip and wrist fractures suffer no significant difference in survivorship and functional outcome at discharge and after 4 months when compared with patients with isolated hip fractures. Our results imply that concomitant wrist fractures in elderly patients with hip fractures do not impact their rehabilitative potential despite the initially more severe trauma. We hypothesize that a lower degree of frailty can explain this subgroup specificity. Prospective studies should assess the potential correlation between rehabilitation efficiency and concomitant hip and wrist fractures, possibly using a frailty index to further investigate the distinctive characteristics of this patient subgroup.

## Figures and Tables

**Table 1 geriatrics-04-00066-t001:** Baseline characteristics of study patients.

	Without Concomitant Wrist Fracture	With Concomitant Wrist Fracture	*p* Value
Number of patients 341	325 (95.3%)	16 (4.7%)	
Gender ^‡^			0.749
Male	67 (20.6%)	2 (12.5%)	
Female	258 (79.4%)	14 (87.5%)	
Age (y.) ^†^	78.17 ± 7.54	78.05 ± 5.48	0.931
Residential status ^‡^			0.132
Home	106 (31.1%)	8 (50.0%)	
Institution	219 (68.9%)	8 (50.0%)	
SPMSQ ^†^	7.22 ± 2.97	7.38 ± 2.75	0.83
Operative risk ^‡^			0.605
ASA 1,2	192 (59.0%)	8 (50.0%)	
ASA 3,4	133 (41.0%)	8 (50.0%)	
Motor FIM preop ^†^	85.02 ± 10.23	83.81 ± 17.53	0.789
IADL preop ^†^	5.39 ± 2.75	5.38 ± 2.39	0.987
Type of fracture ^‡^			0.793
Intracapsular	202 (62.2%)	11 (68.8%)	
Extracapsular	123 (37.8%)	5 (31.2%)	

^†^ The values are given as as the median with interqartile range (IQR); ^‡^ The values are given as the number of patients with the percentage in parentheses; SPMSQ = Short Portable Mental Status Questionnaire, ASA = American Society of Anesthesiologists, FIM = Functional Independence Measure, IADL = Instrumentalized Activities of Daily Living.

**Table 2 geriatrics-04-00066-t002:** Outcomes at discharge.

	Without Concomitant Wrist Fracture	With Concomitant Wrist Fracture	*p* Value
Motor FIM ^†^	42.17 ± 17.89	40.67 ± 12.53	0.748
Complications ^‡^			0.317
Yes	180 (55.4%)	14 (87.5%)	
No	145 (44.6%)	2 (12.5%)	
In-hospital mortality ^‡^			0.332
No	302 (92.9%)	14 (87.5%)	
Yes	23 (7.1%)	2 (12.5%)	
Length of hospital stay ^†^	31.32 ± 18.26	29.44 ± 9.25	0.465

^†^ The values are given as as the median with interqartile range (IQR); ^‡^ The values are given as the number of patients with the percentage in parentheses.

**Table 3 geriatrics-04-00066-t003:** Outcome at 4 months follow-up.

	Without Concomitant Wrist Fracture	With Concomitant Wrist Fracture	*p* Value
Motor FIM gain ^†^	19.34 ± 13.93	17.19 ± 9.31	0.527
IADL ^†^	3.33 ± 2.61	2.58 ± 2.10	0.334
Mortality ^‡^			0.71
No	278 (85.5%)	15 (93.8%)	
Yes	47 (14.5%)	1 (6.2%)	

^†^ The values are given as as the median with interqartile range (IQR); ^‡^ The values are given as the number of patients with the percentage in parentheses.

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
