# Peer review of "Simultaneous Hip and Distal Radius Fractures—Does It Make a Difference with Respect to Rehabilitation?"

_geriatrics, 2019, doi:10.3390/geriatrics4040066_

Round 1

Reviewer 1 Report

This a novel study to examine outcomes for concurrent hip and radial fractures. Small numbers in study with only 4.7% radial and hip fracture rates. Whilst conclusions may be appropriate for this study major limitation is not using a frailty scale to better define the hip fracture population. So I would qualify conclusions because of this

Author Response

Thank you for this valuable comment. We acknowledge the value to use a frailty scale to evaluate the frailty status of our population. Therefore, we stated in our conclusion that 

"Prospective studies should assess the potential correlation between rehabilitation efficiency and concomitant hip and wrist fracture. use a frailty index to further investigate the distinctive characteristics of this patient subgroup."

Reviewer 2 Report

I would like authors to describe why they treated all wrist fractures only with cast? What are their criteria for operative treatment? Do they have different criteria for treating wrist fractures combined with hip fractures vs isolated wrist fractures?

Overall, i find this paper interesting, and i would suggest to continue with this research to reach higher number of patients and maybe longer follow up, maybe up to 1 or even better 2 years.

Author Response

Not all wrist fractures were treated only with a cast. Wrist fractures, which required surgical treatment were excluded from the study. Criteria for operative treatment were based on the AAOS guideline on the treatment of distal radius fractures. No different criteria for wrist fractures were used for treating wrist fractures combined with hip fractures vs isolated wrist fractures. We acknowledged this comment and explained our inclusion criteria more explicitely.